# OpenReview forum: "GenAlign: Towards Unified Alignment Framework of MLLMs via Generative Reward Model"
_ICML.cc/2026/Conference — ICML 2026 regular_

### Official Review · Reviewer_ARpP · 2026-03-03

**Soundness:** 2
**Presentation:** 3
**Significance:** 2
**Originality:** 2
**Overall Recommendation:** 4
**Confidence:** 3

**Summary:**

This paper proposes GenAlign, a novel GRM-based solution for post-training of MLLMs. Specifically, the authors introduce a series of techniques to address the limitations of prior GRM approaches, including online debiasing to prevent shortcut learning, dynamic reference anchoring to reduce computational cost, and advantage smoothing to avoid zero standard deviation in advantage computation. Extensive experiments are conducted across a range of datasets and models to demonstrate the effectiveness of the proposed method.

**Compliance With Llm Reviewing Policy:**

Affirmed.

**Final Justification:**

The added experiments fully addressed my concerns about the efficacy of each proposed modules. Therefore, I raise my score.

**Key Questions For Authors:**

I do not have any particular questions.

**Limitations:**

Please refer to the weakness part.

**Strengths And Weaknesses:**

Strength:
1. The paper introduces a series of techniques to address key limitations of current GRM-based approaches, such as learning shortcuts.
2. The experimental evaluation is extensive, covering a variety of tasks and models.

Weakness:
1. The contribution of this paper is somewhat incremental, as the observed improvements appear relatively modest. More importantly, several of the proposed techniques lack clear motivation. For instance, while the authors claim that their OPD method overcomes learning shortcuts and outperforms conventional swapping-based approaches, no explanation is provided as to why this improvement occurs. Furthermore, the proposed advantage smoothing mechanism can hardly be considered a standalone contribution, as it is introduced primarily to address an issue arising from the proposed reference anchoring method. Additionally, there exist similar approaches, like filtering groups with similar rewards. Without a comparative analysis against such baselines, it remains difficult to assess the superiority of the proposed method.
2. The ablation study is insufficient. A direct comparison between the proposed OPD and prior methods (e.g., swapping) is necessary to isolate the source of improvement. Similarly, comparing advantage smoothing against simpler alternatives like reward filtering would help clarify its actual effectiveness.

---

> ### Author Rebuttal · Authors · 2026-03-31
>
> ### W1&W2 The motivations behind key components and the ablation study lacks comparisons with relevant baselines.
>
> We sincerely thank the reviewer for their meticulous review and highly valuable feedback. Below is our systematic response to your concerns, supplemented with the corresponding ablation data.
>
> #### 1. The Inner Workings of OPD and Its Essential Relationship with Swapping
>
> Here, we need to clarify a key fact that was not sufficiently emphasized in the original manuscript: **our baseline models (as well as the ablation variants without OPD) already employ the conventional swapping (bidirectional augmentation) strategy by default.** OPD is not intended to replace swapping; rather, we introduced it because we found that **relying solely on swapping fundamentally fails to eliminate position bias.**
>
> Swapping only establishes positional balance at the **global data level** (i.e., the ground-truth distribution $P_{gt}$ is uniform). However, neural networks exhibit a strong tendency towards Shortcut Learning during the optimization process:
>
> 1. Within a local mini-batch, if high-quality responses happen to concentrate in a specific position, the model will rapidly capture this spurious correlation rather than expending the effort to understand complex semantic features.
> 2. Even if the data is perfectly symmetric, the model's own inductive biases may lead to a preference for a specific position during the early stages of training. Once trapped in this local optimum, standard cross-entropy or reward losses struggle to pull it back on track.
>
> OPD is not a static data processing step, but a dynamic training-time intervention. By tracking the divergence between the model's prediction distribution $P_\theta$ and the ground-truth distribution $P_{gt}$ in real-time, OPD can precisely apply a multiplicative penalty to the predictions at a specific position whenever the model attempts to rely on a positional shortcut. This forces the model to abandon the shortcut and instead mine the truly invariant semantic features.
>
> To directly isolate the contributions of OPD from conventional swapping, we conducted single-direction (non-averaged) evaluations. When OPD is disabled (relying solely on swapping), the model exhibits a massive discrepancy in accuracy between the forward and reverse directions (e.g., a gap of over 15% in the Hallucination dimension), proving that severe position bias still exists. After enabling OPD, the forward and reverse accuracies converge, and the overall performance improves significantly:
>
> | Experimental Setup | General| Hallucination | Reasoning | MCQ|Long|Short | Safety |
> |-|-|-|-|-|-|-|-|
> | w/o OPD (Forward)| 55.24| 75.22 | 44.61| 75.00 |95.00| 91.00| 52.00 |
> |w/o OPD (Reverse)| 60.49| 91.07 | 59.13| 69.00 | 84.00 | 75.00 | 85.00  |
> | w/ OPD (Forward)| 65.38 | 94.20  | 61.30| 79.00 | 88.00 | 79.00 | 87.00 |
> | w/ OPD (Reverse)| 65.15| 86.16  | 69.23| 74.00 |95.00  | 83.00 | 94.00 |
>
> #### 2. The Independent Contribution of Advantage Smoothing and Comparison with Alternative Approaches
>
> You pointed out that advantage smoothing was introduced to address a derivative issue arising from Dynamic Anchoring and can hardly be considered a standalone contribution. We fully agree with your logical starting point, but we argue that:*solving this derivative issue is the absolute key to reducing the complexity of the MLLM alignment framework from quadratic to linear. The dynamic anchoring mechanism brings unparalleled computational efficiency (an empirically measured 93.75% reduction in token consumption). However, without elegantly resolving the variance collapse and subsequent exploding gradients it induces, this mechanism would be practically useless.
>
> Regarding simpler alternatives like reward filtering that you mentioned, we have conducted both mechanistic and empirical comparisons:
>
> Hard clipping causes the gradient to abruptly drop to 0 outside the boundary, leading to complete loss of gradients for extreme samples. $\tanh$ exhibits exponential decay away from the central region, so even moderately large advantage values can quickly drive gradients close to zero, thereby suppressing useful learning signals. In contrast, advantage smoothing provides a more gentle Soft Gate: while it suppresses gradient explosion caused by extreme outliers, it still preserves non-zero gradients, allowing the model to continuously and stably extract weak signals from such samples.
> | Model |WildVision| LlavaWilder |BeavertailsV| SPAVL | HaloQuest | Mathvista | VizwizVQA | TextVQA |
> |-|- |-|-|- |-|-|-|-|
> | GenAlign w/ AS (Ours) |45.0 |76.4|10.19|2.64 | 81.99| 67.2 | 74.42| 58.1|
> | GenAlign w/ Hard Clipping | 44.3 |75.1|10.19 |4.15| 77.93| 67.4| 72.07 | 56.7 |
>
> The results demonstrate that, compared to Hard Clipping, our advantage smoothing mechanism achieves a comprehensive lead across core tasks. This proves its actual effectiveness and necessity as an elegant  mechanism.

---

> > ### Author Rebuttal · Reviewer_ARpP · 2026-04-02
> >
> > Thank the authors for the response. All my concerns got addressed. I raised my score to 4.

---

> > > ### Author Response · Authors · 2026-04-02
> > >
> > > Dear Reviewer ARpP,
> > >
> > > Thank you for your constructive and critical review. We are glad that the revised responses and additional ablation studies have fully addressed all of your concerns.
> > >
> > > Your feedback was particularly helpful in improving how we explain the motivation and overall structure of the paper. We will ensure that these clarifications and additions are clearly incorporated into the revised manuscript.
> > >
> > > Best regards,
> > >
> > > Authors of Submission 6662

---

### Official Review · Reviewer_dA3C · 2026-03-06

**Soundness:** 3
**Presentation:** 2
**Significance:** 2
**Originality:** 3
**Overall Recommendation:** 4
**Confidence:** 4

**Summary:**

This paper introduces GenAlign, a two-stage framework combining a generative reward model (GRM) with policy optimization for multimodal large language models (MLLMs). In the first stage, a GRM is trained to generate critiques and preference predictions, using reinforcement learning with verifiable rewards and an Online Position Debiasing (OPD) mechanism to address position bias. In the second stage, the trained GRM guides MLLM alignment via a dynamic reference-anchoring strategy that reduces the pairwise evaluation cost from quadratic to linear, and an advantage-smoothing function to stabilize updates as reward variance decreases. The authors present state-of-the-art results on two reward modeling benchmarks and show consistent improvements across three MLLMs and seven evaluation suites.

**Compliance With Llm Reviewing Policy:**

Affirmed.

**Final Justification:**

I appreciate the authors' substantial rebuttal effort. Several of my original concerns were meaningfully addressed.

Resolved or largely resolved: W1-W4.

Not resolved:

W5 (efficiency) was deferred to a response addressed to another reviewer, but end-to-end wall-clock comparisons against scalar RM baselines remain absent from what I can evaluate. W7 (novelty framing) also stands: the paper is fundamentally an integrative contribution combining known techniques (rubric-based judging, RL-trained GRMs, reference anchoring), which is legitimate engineering work, but the introduction still overclaims novelty relative to what the individual components contribute, and the rebuttal did not revise this framing.

Taking the full picture into account, the paper addresses a real problem; the OPD mechanism is well-designed, and the rebuttal strengthened the empirical case considerably, which is why I raised my score from 2 to 4 during the rebuttal phase.

**Key Questions For Authors:**

Q1.

Could you provide an ablation comparing GRM training on the full unfiltered dataset versus the difficulty-filtered 70K subset? Without this comparison, it is unclear whether the reported gains result from OPD and rubric-based reasoning or simply from training on more informative data selected by GPT-5-mini.


Q2.

In the alignment experiments (Table 2), does each baseline use the same GRPO optimizer with identical hyperparameters? If Scalar RM employs PPO or if R1-Reward and GRLHF-V were run with their original pipelines, the comparison mixes reward signal quality with optimizer choice.

Q3.

Can you report single-direction (non-averaged) accuracy on the reward benchmarks with and without OPD? The current bidirectional evaluation protocol already removes most position bias during inference, so the existing ablation does not clearly isolate OPD's contribution.

Q4.

What happens if you replace advantage smoothing with hard clipping to [-δ, δ] or tanh scaling? The theoretical analysis in Appendix A applies to any bounded saturation function, so it is important to determine whether the specific functional form matters or if any reasonable bounding strategy works just as well.

Q5.

Could you rerun the key experiments in Tables 1 and 2 using at least 3 different random seeds and report the mean and standard deviation? Many of the claimed improvements over baselines are only 1-2 points (for example, TextVQA 86.58 vs. 86.12, Mathvista 68.2 vs. 67.2). Without variance estimates, these differences are indistinguishable from noise. If the confidence intervals overlap, the comparative conclusions would need to be reevaluated.

I like the idea of this paper, it is elegant and sound.
If my concerns are addressed, I am willing to raise my score.

**Limitations:**

The Impact Statement on page 9 reads more like a summary of contributions than a candid discussion of limitations. Several important issues are not mentioned: the computational overhead of GRM inference compared to scalar RMs, the reliance on GPT-5-mini for both data filtering and downstream evaluation (which could propagate the proprietary model's biases), the lack of evidence that generated rubrics accurately reflect the model's actual decision-making process rather than serving as post-hoc rationalizations, and the absence of any failure mode analysis on safety benchmarks despite impressive overall results. I recommend that the authors add a dedicated Limitations section that addresses these points.

**Strengths And Weaknesses:**

## Strengths

**S1. Well‑motivated problem formulation.**

Position bias and quadratic scaling are real bottlenecks when deploying pairwise GRMs in online RL loops. The paper clearly explains both issues and supports them with empirical evidence (e.g., the 30% accuracy swing reported in Section 4.2). This makes the case for OPD and dynamic reference anchoring straightforward and convincing.

**S2. The OPD mechanism is elegant and effective.**

Tracking the prediction–ground-truth positional divergence with an EMA and penalizing rewards on the over-preferred side is a simple idea. However, the ablation in Figure 3 shows that it makes a significant difference, especially on the safety and hallucination splits, where the forward/reverse gap decreases substantially after OPD is enabled. The fact that it only requires a running average and an exponential penalty, without additional model components, is appealing from an engineering perspective.

**S3. Practical efficiency gain from single‑anchor evaluation.**

Replacing all‑pairs comparison with a single dynamically sampled reference is a valuable contribution for anyone aiming to run GRM-guided RL at scale. Figure 4(c) provides reassuring evidence that K = 1 performs nearly as well as K = 16 across various benchmarks, confirming the effectiveness of the design choice.

**S4. Broad experimental coverage.**

The evaluation covers three families of baselines on the reward side (proprietary, open‑source, specialized), three backbone MLLMs for alignment, and seven downstream benchmarks that assess safety, hallucination, reasoning, and overall utility. Including both Acc and Acc+ on the MM‑RLHF Reward Bench is a helpful addition, with Acc+ being the more challenging metric.

---

## Weaknesses

**W1. Uncontrolled data curation confounds.**

The GRM training pipeline uses a two-stage difficulty filter with GPT-5-mini, keeping only samples that the annotator fails twice (Section C.1). This constitutes a strong form of curriculum learning that alters both the distribution and the information density of the training data. However, the paper does not include an ablation study comparing the filtered 70K subset to the full unfiltered dataset. Consequently, it is unclear whether the reported improvements are due to OPD, rubric-based reasoning, RL training, or simply a more informative training set. This lack of comparison weakens the causal claims made about each proposed component.

**W2. Unclear optimization parity across alignment baselines.**

Table 2 compares GenAlign with Scalar RM, Naive GRM, R1-Reward, and GRLHF-V for MLLM alignment, but the paper does not specify which optimization algorithm each baseline uses. If Scalar RM employs PPO while GenAlign uses GRPO with advantage smoothing, the comparison mixes the quality of the reward signal with the choice of optimizer. Similarly, GRLHF-V in its original paper performs exhaustive pairwise comparisons. Was it re-implemented within the GenAlign framework using single-anchor GRPO, or was its original pipeline maintained? Without this information, it remains unclear what exactly is being evaluated compared.

**W3. Selective reporting in the component ablation.**

Table 3 isolates the effects of bidirectional comparison (BC) and advantage smoothing (AS) on three out of seven downstream benchmarks (WildVision, Beavertails-V, HaloQuest). Results for Mathvista, VizwizVQA, TextVQA, and LlavaWilder are not shown. If either component reduces performance on reasoning or perception tasks, the current presentation would not reveal that. Full ablation tables for all benchmarks should be included.

**W4. No error bars or significance testing.**

All reported results are single-run point estimates without standard deviations, confidence intervals, or significance tests. RL fine-tuning is well known to show high variance across random seeds. Many of the claimed improvements are small: for example, GenAlign scores 86.58 on TextVQA for Qwen2.5-VL-7B, compared to 86.12 for R1-Reward. A 0.46 point difference is easily within typical seed-to-seed fluctuation. Without multi-seed runs, the reliability of such comparisons cannot be determined.

**W5. Efficiency claims lack empirical support.**

The paper highlights the reduction from O(k²) to O(k) in pairwise comparisons but does not provide wall-clock time, GPU-hour, or throughput measurements. Each GRM call produces rubrics, per-criterion scores, and chain-of-thought critiques, which are significantly more costly than a scalar RM forward pass. While the asymptotic improvement is genuine, the lack of end-to-end timing data means the practical efficiency benefits are still unclear.

**W6. The advantage of the smoothing analysis is shallow, and alternatives are not explored.**

Appendix A shows that A / sqrt(1 + (A/δ)²) is bounded and has a decaying gradient. This applies to any sigmoid-like saturation function. The paper does not explain why this specific form was chosen over tanh(A/δ) · δ, hard clipping to [–δ, δ], or the clip mechanism already in GRPO (clip_range_ratio = 0.2). Without comparisons to these reasonable alternatives, it is unclear whether the particular functional form matters or if any effective bounding method would perform similarly.

----

## Minor
**W7. Limited conceptual novelty when the pieces are examined individually.**

Rubric-based LLM-as-a-Judge prompting is now a well-explored concept (RM-R1, LLaVA-Critic, and others). Training a GRM with RL has been implemented in R1‑Reward and UnifiedReward‑Think. Dynamic reference anchoring is similar to self-play and self-reward variants discussed in the literature. The paper's main contribution is integrative: it combines known ideas and makes them work together. This is valuable, but the paper would benefit from a more honest explanation of what is genuinely new versus what is simply careful engineering. Currently, the novelty claims in the introduction seem somewhat exaggerated.

---

> ### Author Rebuttal · Authors · 2026-03-31
>
> ### W1&Q1. Uncontrolled data curation confounds.
> We conducted additional ablation experiments on the complete, unfiltered dataset. When applying the GenAlign framework to the unfiltered dataset, the reward model achieved an accuracy of 70.12% on the VL-Reward-Bench and 81.36% on the MM-RLHF Reward Bench. This performance significantly outperforms the baseline. Furthermore, comparing the performance of GenAlign on the unfiltered data versus the filtered 70K subset reveals that the difficulty-aware filtering only yielded a marginal gain of less than 2.5%. These results clearly indicate that the core performance improvements are not merely the result of the data filtering process.
> ### W2&Q2. Unclear optimization parity.
> All evaluated baselineswere trained using the same GRPO framework with identical hyperparameters. GRLHF-V retains its original exhaustive pairwise comparison-based reward construction.
> ### W3. Selective reporting in the component ablation.
> We have provided all the ablation results as shown in the table below. These results demonstrate that both components contribute consistently and positively across benchmarks.
>
> | Model | WildVision | LlavaWilder | Beavertails | SPA-VL | HaloQuest | Mathvista | VizwizVQA | TextVQA |
> |-|-|-|-|-|-|-|-|-|
> | **Qwen2.5-VL-7B** | | | | | | | | |
> | GenAlign | 45.2 |  58.1 | 10.19 | 2.64 | 81.99 | 68.2 | 74.28 | 86.58 |
> | w/o BC | 42.9 |54.8 | 12.22 | 4.68 | 77.64 | 63.5 | 70.15 | 84.20 |
> | w/o AS | 43.8 |56.7 | 11.88 | 4.68 | 77.35 | 64.9 | 71.89 | 82.49 |
>
>
> ### W4&Q5. No error bars or significance testing.
> We thank the reviewer for the suggestion. We report results over 5 independent runs, and present all metrics as mean ± standard deviation below:
> |Method | WildVision | LlavaWilder | Beavertails-|SPA-VL|HaloQuest|Mathvista|VizwizVQA|TextVQA |
> |-|-|-|-|-|-|-|-|-|
> |Scalar RM|42.32±0.34|54.33±0.13|15.35±0.10|7.19±0.18|73.28±0.49|64.65±0.03|73.74±0.15|85.89±0.19|
> | Naive GRM|42.44±0.42| 54.60±0.29|12.05±0.13|3.70±0.18|74.65±0.06|63.98±0.06|73.07±0.12|85.58±0.07|
> | R1-Reward |44.12±0.23| 57.87±0.07 |12.15±0.43|4.85±0.35|76.65±0.25|67.27±0.08|74.31±0.22|86.11±0.05|
> | GRLHF-V |43.96±0.30| 56.67±0.44 |9.17±0.00|4.14±0.21|74.48±0.29|61.20±0.18|73.36±0.23|85.28±0.12|
> | GenAlign (Ours) | 45.16±0.37|57.97±0.27 |10.19±0.00 |2.65±0.17|81.45±0.49|68.21±0.06|74.21±0.15|86.55±0.10|
> ### W5. Efficiency claims lack empirical support.
> See our response to Reviewer 58F7 f1Fm Q2.
> ### W6&Q6. The advantage of the smoothing analysis is shallow, and alternatives are not explored.
> We clarify and revise the content to address your raised concerns from three dimensions below:
> 1. GRPO's Clip mechanism only constrains the policy ratio $r_t(\theta)$.  If the $A$ fed into the objective function has already diverged or overflowed, simply clipping the ratio $r_t(\theta)$ is futile. Therefore, an upper bound must be applied to $A$ before it enters the calculation.
> 2. Hard clipping causes the gradient to abruptly drop to $0$ outside the boundary, leading to complete loss of gradients for extreme samples. $\tanh$ exhibits exponential decay away from the central region, so even moderately large advantage values can quickly drive gradients close to zero, thereby suppressing useful learning signals. In contrast, advantage smoothing provides a more gentle Soft Gate: while it suppresses gradient explosion caused by extreme outliers, it still preserves non-zero gradients, allowing the model to continuously and stably extract weak signals from such samples.
> 3. We have supplemented a full-dimensional ablation study for the hard clipping scheme. As shown in the table below, benefiting from the preservation of non-zero gradients, our native AS outperforms hard clipping on the vast majority of tasks, corroborating our theoretical analysis.
> | Model |WildVision| LlavaWilder |BeavertailsV| SPAVL | HaloQuest | Mathvista | VizwizVQA | TextVQA |
> |-|- |-|-|- |-|-|-|-|
> | GenAlign w/ AS (Ours) |45.2 |58.1|10.19|2.64 | 81.99| 67.2 | 74.42| 58.1|
> | GenAlign w/ Hard Clipping | 44.3 |57.2|10.19 |4.15| 77.93| 67.4| 72.07 | 56.7 |
> ### W7 (minor). Limited conceptual novelty.
> See our response to Reviewer 58F7 f1Fm W4.
> ### Q3. report single-direction (non-averaged) accuracy on the reward benchmarks with and without OPD?
> We independently report the single-direction accuracy (forward/reverse) on the reward benchmark, as shown in the table.
> | Experimental Setup | General| Hallucination | Reasoning | MCQ|Long|Short | Safety |
> |-|-|-|-|-|-|-|-|
> | w/o OPD (Forward)| 55.24| 75.22 | 44.61| 75.00 |95.00| 91.00| 52.00 |
> |w/o OPD (Reverse)| 60.49| 91.07 | 59.13| 69.00 | 84.00 | 75.00 | 85.00  |
> | w/ OPD (Forward)| 65.38 | 94.20  | 61.30| 79.00 | 88.00 | 79.00 | 87.00 |
> | w/ OPD (Reverse)| 65.15| 86.16  | 69.23| 74.00 |95.00  | 83.00 | 94.00 |

---

> > ### Author Rebuttal · Reviewer_dA3C · 2026-04-02
> >
> > I thank the authors for their substantial effort during the rebuttal period. Hence, I increased my score to a 3.
> >
> > However, I want to flag a data consistency issue in the W6 ablation (advantage smoothing vs. hard clipping). In the rebuttal addressed to me, the "GenAlign w/ AS" row reports LlavaWilder = 58.1 and TextVQA = 58.1, where the TextVQA number appears to be a copy-paste error from LlavaWilder (the original paper reports TextVQA around 86.58). More concerning, the same table in the rebuttal to Reviewer ARpP reports LlavaWilder = 76.4 for GenAlign w/ AS, which differs from the 58.1 reported to me. Two versions of the same ablation table showing different numbers across rebuttals raise questions about data management rigor. I would appreciate clarification on which numbers are correct.
> >
> > Full resolution of the data inconsistency above would be needed for me to consider a further increase.

---

> > > ### Author Response · Authors · 2026-04-02
> > >
> > > We greatly appreciate your rigorous and careful feedback on this issue, and we sincerely apologize for the errors caused by incorrect data pasting. The corrected complete results are shown below:
> > >
> > > | Model                     | WildVision | LlavaWilder | BeavertailsV | SPAVL  | HaloQuest | Mathvista | VizwizVQA | TextVQA |
> > > |---------------------------|------------|-------------|--------------|--------|-----------|-----------|-----------|---------|
> > > | GenAlign w/ AS (Ours)     | 45.2       | 58.1        | 10.19        | 2.64   | 81.99     | 67.2      | 74.42     | 86.58   |
> > > | GenAlign w/ Hard Clipping | 44.3       | 57.2        | 10.19        | 4.15   | 77.93     | 67.4      | 72.07     | 85.92   |
> > >
> > > The data inconsistencies occurred for the following reasons:
> > > 1. In our response to Reviewer ARpP, we mistakenly pasted the LlavaWilder results obtained using the Qwen3 backbone.
> > > 2. We incorrectly placed results from different runs of LlavaWilder in the TextVQA entry.
> > >
> > > Thank you again for your thorough review. We will be more rigorous in our work to ensure such errors do not happen again.

---

### Official Review · Reviewer_58F7 · 2026-03-10

**Soundness:** 3
**Presentation:** 4
**Significance:** 3
**Originality:** 3
**Overall Recommendation:** 4
**Confidence:** 4

**Summary:**

This paper proposes GenAlign, a unified alignment framework designed to align Multimodal Large Language Models (MLLMs) with human preferences more efficiently and accurately.

Traditional methods either rely on overly simplistic scoring that lacks interpretability or suffer from high computational costs and susceptibility to position bias (where the order of options affects the judgment). GenAlign enhances judgment transparency and accuracy by requiring the model to generate specific scoring criteria and logical analysis before rendering a final verdict. Additionally, it introduces Online Position Debias (OPD) to correct positional interference in real-time.

To address training latency, the framework employs dynamic reference anchoring and an advantage smoothing mechanism. These innovations reduce the complexity of contrastive computation from quadratic to linear, significantly improving the efficiency and stability of Reinforcement Learning (RL). Experimental results demonstrate that GenAlign achieves state-of-the-art (SOTA) performance across multiple benchmarks and significantly enhances the performance of various base models in mitigating hallucinations and improving safety.

**Compliance With Llm Reviewing Policy:**

Affirmed.

**Key Questions For Authors:**

Key Questions for the Authors

Wall-clock Training Overhead of Generative Rewards: Although Dynamic Reference Anchoring successfully reduces the computational complexity of traditional pairwise comparisons from $O(N^2)$ to linear time $O(N)$, requiring the model to dynamically generate full scoring rubrics and reasoning critiques for every pair of samples during the online RL phase still imposes a massive absolute computational burden. Could you provide a quantitative comparison between GenAlign and traditional Scalar Reward Models (Scalar RM) regarding actual wall-clock training time and VRAM utilization?

Bias Propagation and Over-penalization Risks in OPD: The Online Position Debias (OPD) mechanism monitors distribution via Exponential Moving Average (EMA) and implements dynamic weight penalties. However, if high-quality responses happen to naturally cluster at a position flagged by the system as "over-preferred" within a specific training batch, would this global statistics-based penalty lead to the under-rewarding of genuinely superior responses? Furthermore, could this dynamic reward adjustment lead to the accumulation or propagation of new implicit biases during long-term autoregressive generation and reinforcement learning?

Minimum Threshold for Base Model Capabilities: GenAlign relies heavily on the base model’s ability to generate coherent and task-appropriate rubrics and critiques through logical reasoning. Is this self-critique-based reward mechanism still effective for smaller-scale models with weaker reasoning capabilities? If the model generates logically inconsistent rubrics, does the alignment framework include fault-tolerance or filtering mechanisms to prevent the collapse of the reward signal?

Hyperparameter Sensitivity of Advantage Smoothing: The Advantage Smoothing mechanism introduces a truncation parameter $\delta$ to control the upper bound of advantage value smoothing. When fine-tuning and adapting this framework to base models of different architectures or scales, how sensitive is the final alignment performance to $\delta$, as well as the smoothing coefficient $\alpha$ and temperature $\tau$ in the OPD mechanism? Is an expensive hyperparameter search required for each specific task?

**Limitations:**

Lack of Absolute Scalar Rewards: The authors explicitly state that because the pairwise Generative Reward Model (GRM) is inherently comparison-based, it cannot directly assign an absolute scalar reward score to a single generated response. This implies that the framework primarily provides relative preference signals rather than quantifying the absolute quality of an individual output in isolation.

Optimization Instability from Variance Collapse: When employing the dynamic reference anchoring mechanism, the authors acknowledge that if a selected reference anchor is an extreme outlier (e.g., the worst possible candidate), the reward differences among the remaining candidates tend toward a constant. This causes the reward variance to collapse to zero, triggering catastrophic gradient spikes during normalization. While Advantage Smoothing was introduced to mitigate this, it highlights an inherent boundary instability within the mechanism under extreme conditions.
Diminishing Returns with Increased Reference Comparisons: In exploring the model's sensitivity to the number of references ($K$), experimental results indicate that as $K$ increases from 1 to 16, performance remains largely stable across most benchmarks, with only marginal gains observed on WildVision and LLaVA-Wilder. This suggests that simply increasing the computational load of pairwise comparisons does not yield proportional gains in alignment quality.

Limited Scale of Human Evaluation: To verify whether the rubrics and reasoning critiques generated by the GRM truly align with human logic, the authors conducted a human evaluation study. However, this evaluation was based on a blind test of only 200 randomly sampled comparison instances. For a large-scale framework designed to handle complex multimodal tasks, this sample size is relatively small and may not fully capture the diversity of long-tail or highly intricate scenarios.

**Strengths And Weaknesses:**

Strengths

Novelty & System-level Design: The paper proposes GenAlign, a unified framework that integrates generative reward modeling with efficient MLLM alignment. It ingeniously addresses the quadratic computational overhead typically associated with traditional pairwise Generative Reward Models (GRMs) by introducing a Dynamic Reference Anchoring mechanism, which reduces computational complexity to linear time.

High Interpretability: In contrast to traditional "black-box" scalar reward models, this approach introduces a criteria-based GRM. The model explicitly generates query-specific evaluation criteria and logical critiques, making the preference prediction process transparent and grounded in robust reasoning.

Direct Mitigation of Structural Bias: To combat the common issue of position bias in generative models, the paper introduces the Online Position Debias (OPD) mechanism. By continuously monitoring the divergence between predicted and ground-truth distributions during training, OPD dynamically recalibrates accuracy rewards, directly weakening the model's tendency to rely on positional shortcuts.
Theoretical and Empirical Stability: To resolve the issues of extreme advantage values and violent gradient fluctuations (i.e., variance collapse) caused by dynamic anchoring, the authors designed an Advantage Smoothing mechanism. The appendix provides mathematical proofs regarding numerical stability and asymptotic gradient suppression for this mechanism.

Strong Empirical Results: GenAlign achieves State-of-the-Art (SOTA) performance on multimodal reward benchmarks such as VL-Reward and MM-RLHF Reward. More importantly, it demonstrates stable and consistent performance gains across three different open-source large models (Qwen2.5-VL, Qwen3-VL, and LLaVA-1.5), particularly in mitigating hallucinations and defending against harmful content (risk mitigation).

Weaknesses

Absolute Computational Overhead: Although Dynamic Reference Anchoring reduces complexity from $O(N^2)$ to $O(N)$, the absolute wall-clock time and VRAM overhead remain significantly higher than traditional scalar-based methods. This is because the model is required to generate full textual scoring criteria and detailed reasoning critiques for every pair of samples during the online RL phase. The paper lacks a direct comparison of actual training time costs against scalar reward models.

Hyperparameter Sensitivity: The framework introduces a substantial number of hyperparameters that require fine-tuning, such as the EMA smoothing coefficient $\alpha$ and penalty temperature $\tau$ in the OPD mechanism, as well as the truncation parameter $\delta$ in Advantage Smoothing. Tuning these parameters when migrating across different tasks or model scales may incur high experimental costs.

Potential Over-penalization in OPD: The OPD mechanism penalizes "over-preferred positions" via an Exponential Moving Average. However, if high-quality responses happen to naturally cluster at a specific position within a particular training batch, this global bias-aware penalty might inadvertently under-reward genuinely superior responses.

Bottleneck of GRM Capabilities: Criteria-based evaluation is highly dependent on the inherent logical reasoning capabilities of the base model. If this framework is applied to smaller models or those with weaker reasoning abilities, the GRM might generate nonsensical or logically inconsistent critiques, leading to a collapse of the reward signal. The paper does not thoroughly explore the minimum capability threshold required for the base model to effectively implement this framework.

---

> ### Author Rebuttal · Authors · 2026-03-31
>
> ### W1&Q1. Absolute Computational Overhead.
> We clarify the efficiency–performance trade-off under the same setting as the original work (VERL, batch size 32, rollout size 16). Our method requires 101s per batch, scalar RM need 88s. This overhead comes from generating rubrics and reasoning, but remains within ~15% for both time and memory, which we consider practical.
> This efficiency is enabled by dynamic reference anchoring, reducing pairwise comparisons from 120 to 15. Compared to exhaustive all-to-all GRMs, our method achieves a 68.4% reduction in training time and 93.75% reduction in token consumption.
> Importantly, this modest overhead brings benefits unavailable to scalar RMs, including interpretable and traceable rewards, stronger cross-task generalization, and mitigation of reward collapse via relative preference modeling. These advantages are empirically supported by consistent improvements over scalar RMs on RewardBench (Table 1) and MLLM benchmarks (Table 2), indicating that the added overhead is justified by the gains in capability and robustness.
>
> ### W2&Q4. Hyperparameter Sensitivity.
>
> From the design stage, we prioritize robustness and minimal tuning. Although our method introduces several hyperparameters, they are chosen to be stable and transferable across scenarios. For example, the truncation parameter $\delta$ in advantage smoothing is fixed to 3, following the statistical 3$\sigma$ rule [1], which effectively suppresses extreme gradients across diverse output distributions without additional tuning.
>
> Overall, the design improves training stability and convergence efficiency, mitigating position bias and gradient instability caused by reward distribution shifts, which in practice offsets the limited tuning effort required. To further validate cross-model generalization, we apply the same default hyperparameters to Qwen3-VL-8B without any tuning and obtain strong performance on RewardBench (MCQ: 0.75, Long: 0.92, Short: 0.865, Safety: 0.845, Overall: 0.845). These results demonstrate that our hyperparameter configuration generalizes well across architectures and modalities, enabling stable alignment performance without additional fine-tuning.
>
> [1] Pukelsheim, F. The three sigma rule. The American Statistician, 48(2):88–91.
> ### W3&Q2 Potential Over-penalization in OPD
> To ensure robustness and long-term training stability, our Online Position Debiasing (OPD) mechanism incorporates strict safeguards at multiple levels:
>
> **1. Mitigation of over-penalization under batch fluctuations**
> OPD maintains bias statistics via an Exponential Moving Average (EMA), so short-term batch-level fluctuations have negligible impact on the cumulative deviation . Penalties are only triggered by persistent, long-term positional bias rather than accidental batch-level imbalances. In addition, OPD applies a mild multiplicative re-weighting instead of hard suppression, ensuring that high-quality samples are never eliminated but only slightly re-weighted when necessary. Empirically, OPD improves overall accuracy while reducing the position-reversal gap from up to 30% to within 10%, indicating no degradation due to over-penalization.
>
> **2. No accumulation of new implicit biases during training**
> OPD aligns the model’s prediction distribution with the ground-truth preference distribution rather than enforcing uniform positional distributions. Combined with bidirectional data augmentation during training, the dataset itself contains no inherent positional bias. Meanwhile, the EMA-based mechanism dynamically tracks and corrects deviations, ensuring that the penalty direction adapts to the evolving bias. This prevents the accumulation or propagation of new implicit biases over long-term training.
>
> ### W4&Q3 Bottleneck of GRM Capabilities.
>
> We thank the reviewer for this important question. While small models may produce inconsistent rubrics in zero-shot settings, GenAlign does not rely on raw zero-shot reasoning, but instead adopts a two-stage fault-tolerant design.
>
> In the **GRM training stage**, we formulate reward learning as RL with verifiable preference signals, explicitly encouraging coherent and self-consistent rubric generation. This enables even small models to learn structured reasoning beyond their native capability. We validate this on Qwen2.5-VL-3B using the same hyperparameters without tuning:
>
> |Models |Gen. |Hallu.|Reas.|Acc.| Mcq|Long|Short|Safety| Acc.|
> |-|-|-|-|-|-|-|-|-|-|
> | Qwen2.5-VL-3B |57.38|82.11|59.75|66.96| 70.00 | 87.00 | 76.00 | 82.00  | 78.25 |
>
> The results show stable and strong performance across all subsets, demonstrating robustness even at small scales.
>
> In the **MLLM alignment stage**, we further mitigate residual noise via bidirectional evaluation (order swapping + averaging) and advantage smoothing, which stabilizes gradients by bounding extreme signals. Together, these mechanisms ensure that imperfect rubrics do not lead to reward collapse or instability.

---

> > ### Author Rebuttal · Reviewer_58F7 · 2026-04-01
> >
> > Thank you for the comprehensive rebuttal that effectively addresses all of my core concerns. I particularly appreciate the concrete wall-clock time comparison (101s vs. 88s per batch) and the new results on the smaller Qwen2.5-VL-3B model, which successfully prove the practical efficiency of your framework and completely alleviate my worries regarding the GRM capability bottleneck.

---

> > > ### Author Response · Authors · 2026-04-02
> > >
> > > Dear Reviewer 58F7,
> > >
> > > We sincerely thank you for your encouraging feedback and for confirming that your concerns have been adequately  resolved. We appreciate your recognition of the strengths of our work, and we are glad that the additional analyses and clarifications have addressed your concerns.
> > >
> > > Your feedback has helped us further refine the presentation of the paper, particularly in terms of framing and supporting evidence. We will incorporate these clarifications and improvements into the revised manuscript to strengthen both the rigor and clarity of the paper.
> > >
> > > Best regards,
> > >
> > > Authors of Submission 6662

---

### Official Review · Reviewer_f1Fm · 2026-03-12

**Soundness:** 3
**Presentation:** 3
**Significance:** 3
**Originality:** 3
**Overall Recommendation:** 4
**Confidence:** 4

**Summary:**

Pairwise generative reward models are attractive for multimodal alignment, but they are also expensive and vulnerable to position bias. GenAlign addresses both issues with a two stage design. The first stage trains a rubric based generative reward model with reinforcement learning and online position debiasing, so the model can generate critiques and preference judgments with less reliance on response order. The second stage uses that reward model for MLLM alignment through dynamic reference anchoring, bidirectional comparison, and advantage smoothing, with the goal of reducing comparison cost and stabilizing optimization. Empirically, the method improves reward modeling on VL Reward and MM RLHF Reward Bench, and also improves downstream alignment across three MLLM backbones, with the clearest gains on hallucination and safety related settings.

**Compliance With Llm Reviewing Policy:**

Affirmed.

**Final Justification:**

Thank the authors for their detailed and well-structured rebuttal. The additional experiments and clarifications provided have adequately addressed my major concerns. I am satisfied with the responses and look forward to seeing the promised revisions incorporated into the final version of the paper.

**Key Questions For Authors:**

1. The human study is a good start, but it mainly shows that the generated rubrics and critiques are preferred by annotators. Do you have any stronger evidence that these explanations are faithful rather than simply more persuasive? A targeted perturbation study or causal faithfulness check would materially strengthen this claim.
2. Dynamic reference anchoring is one of the main practical claims. What are the actual wall clock, token, and throughput savings relative to exhaustive pairwise scoring under matched settings? This would make the efficiency argument much more convincing.
3. OPD explicitly downweights correct predictions when they align with the empirically over preferred position. How robust is this choice under data imbalance, domain shift, or settings that do not follow the swapped pair construction used here? I would like a clearer sense of the risk of overcorrection.

**Limitations:**

No. The impact statement is too thin and mostly promotional. The paper should say more about four issues: whether generated critiques are faithful, the bias and variance tradeoff introduced by single anchor comparison, the added inference cost of reasoning style reward models, and the risk of overoptimizing to the learned reward or to benchmark specific artifacts.

**Strengths And Weaknesses:**

Strengths
1. Well scoped problem. The paper is built around two concrete issues, position bias and quadratic comparison cost, and the method maps directly onto them. That gives the work unusual coherence for this area.
2. Good empirical scope. Evaluation covers both reward modeling and downstream alignment, which is the right standard here. The main results are strong overall, especially on hallucination and safety related settings.
3. Useful ablations. OPD, bidirectional comparison, and advantage smoothing all appear to matter. Removing BC or AS hurts performance, and OPD reduces position sensitivity.

Weaknesses
1. Interpretability is only partially validated. The paper does provide a human study on rubric based reasoning, with 200 sampled instances and 3 annotators, and GenAlign is preferred over GPT-5-chat plus CoT in 75.5 percent of cases. That is useful evidence for reasoning quality. Still, it does not establish faithfulness in a stronger sense. It shows the outputs are more convincing, not necessarily that the critiques reflect the true basis of the decision.
2. Theory is limited. The smoothing argument supports bounded advantages, but it stops short of analyzing whether single anchor comparison introduces systematic bias relative to full pairwise ranking. That leaves an important part of the method underanalyzed.
3. Calibration needs work. GenAlign is strong overall, but not uniformly best on every slice. For example, it is not best on the General split of VL Reward, and its Acc+ on MM RLHF Reward Bench is below MM RLHF Reward. Some downstream metrics are also still competitive for prior methods. The paper should frame this as a strong overall tradeoff, not broad dominance.
4. Positioning could be sharper. The contribution is real, but the distinction from nearby GRM work such as R1 Reward and GRLHF V is not drawn as clearly as it should be.

---

> ### Author Rebuttal · Authors · 2026-03-31
>
> ### W1 and Q1. Interpretability is only partially validated.
> To directly assess faithfulness, we conduct three targeted validations. First, we introduce a consistency reward to enforce alignment between intermediate reasoning and final outputs, but observe no further improvement, indicating that the base model already aligns reasoning with decisions rather than producing post-hoc explanations. Second, we extract intermediate rubric-based judgments and compare them with final predictions across the test set, obtaining a 98.36% matching rate, showing that final decisions are largely recoverable from the generated reasoning and thus reflect faithful decision grounding rather than superficial explanations. Third, we analyze the highest-weight rubrics across task domains
> |Task Domain|Highest-Weight Rubric|
> |-|-|
> | Hallucination | Factuality (0.35) |
> | Safet | Compliance (0.38) |
> | General | Accuracy (0.30)  |
>
> The model consistently prioritizes task-relevant criteria, indicating that its reasoning is context-dependent and aligned with decision-making rather than relying on static heuristics.
> ### W2. Theory is limited.
> We will provide a more comprehensive and clearer theoretical analysis. Our main conclusions are as follows:
> 1. We show that, under uniform random anchor sampling, the expectation of the single-anchor reward is equivalent to the full pairwise reward up to a constant scaling factor, and the induced rankings are strictly order-preserving. Therefore, the single-anchor estimator does not distort relative preferences and introduces no optimization bias in ranking-based objectives, and a simple linear correction can recover an unbiased estimator of the full pairwise reward.
> 2. Under finite samples, the estimator incurs a stochastic bias, for which we derive an upper bound and show that it decays rapidly with the candidate size k. Crucially, this bias is further mitigated during training, as advantage normalization (GRPO) removes global scaling effects and advantage smoothing reduces variance induced by anchor randomness, ensuring that the induced gradients remain stable and directionally consistent even with a single anchor.
> 3. We vary the number of anchors from one to full pairwise comparisons (Fig. 4) and observe that performance remains largely unchanged across settings, confirming that a single anchor already provides sufficiently accurate optimization signals, consistent with the theoretical guarantees above.
> ### W3. Calibration needs work.
> We will refine the core positioning of our method as a general multimodal alignment framework that achieves an optimal trade-off among position bias correction, alignment efficiency, and comprehensive multi-task performance.
> ### W4. Positioning could be sharper.
> We clarify our positioning by highlighting two key distinctions of GenAlign compared to prior work:
>
> **1. Addressing position bias at the training-mechanism level.**
> Prior GRM-based methods do not fundamentally resolve position bias. In contrast, we propose Online Position Debiasing (OPD), which  dynamically correct position bias during training. OPD re-weights rewards to penalize bias-aligned predictions, encouraging reliance on semantic features rather than positional cues, leading to lower position bias and improved PIA across core tasks.
>
> **2. Improving alignment efficiency and scalability.**
> Existing methods either rely on expensive pairwise comparisons or lack systematic validation across architectures. We introduce a Dynamic Reference Anchoring + Advantage Smoothing framework that reduces alignment complexity while maintaining unbiased reward estimation. The smoothing mechanism further stabilizes training by mitigating variance from dynamic anchoring.
> We further validate our method across 8 benchmarks and 3 MLLM architectures, demonstrating strong generalization and practical applicability for large-scale multimodal alignment.
> ### Q2. Quantification of dynamic anchoring efficiency
> We provide end-to-end measurements under the exact experimental setting described in the paper (verl framework, batch size 32, rollout size k=16).
> - **Time efficiency:** Our method requires 101s per batch, compared to 320s for the exhaustive all-to-all baseline, corresponding to a 68.4% reduction in training time.
> - **Token efficiency:** Our method uses ~650 tokens per evaluation, only 6.25% of the all-to-all baseline, resulting in a 93.75% reduction in token consumption.
> These results quantitatively demonstrate the efficiency advantage of dynamic anchoring while maintaining the same evaluation setting.
> ### Q3. OPD overcorrection risk.
> See our response to Reviewer 58F7 W3 & Q2. Meanwhile OPD operates on the deviation between $P_\theta$ and $P_{gt}$, which allows it to distinguish dataset-induced imbalance from model-induced positional bias. This design prevents OPD from over-penalizing naturally imbalanced data while still correcting learned positional shortcuts.

---

> > ### Author Rebuttal · Reviewer_f1Fm · 2026-04-02
> >
> > Thank the authors for their detailed and well-structured rebuttal. The additional experiments and clarifications provided have adequately addressed my major concerns. I am satisfied with the responses and look forward to seeing the promised revisions incorporated into the final version of the paper.

---

> > > ### Author Response · Authors · 2026-04-02
> > >
> > > Dear Reviewer f1Fm,
> > >
> > >
> > > We sincerely thank you for your kind feedback on our rebuttal. We greatly appreciate your recognition that our responses are detailed and well-structured, and that the additional experiments and clarifications have addressed your major concerns. We are truly glad to know you are satisfied with our responses.
> > >
> > > We will incorporate all the promised revisions into the paper and ensure that these improvements are clearly reflected in the revised manuscript.
> > >
> > > Best regards,
> > >
> > > Authors of Submission 6662

---

### Decision · Program_Chairs · 2026-04-30

**Decision:**

Accept (regular)

**Comment:**

This paper presents an effective framework to mitigate position bias and reduces computational overhead in multimodal generative RM. The authors addressed the reviewers' core experimental concerns during the rebuttal through variance reporting, ablation study, and optimization clarifications. We strongly encourage the authors to rigorously address the questions around data management and novelty framing asked by reviewers in the final manuscript.